# Polychlorinated Dibenzo-p-Dioxins, Polychlorinated Dibenzofurans, and Dioxin-Like Polychlorinated Biphenyls in Umbilical Cord Serum from Pregnant Women Living Near a Chemical Plant in Tianjin, China

**DOI:** 10.3390/ijerph16122178

**Published:** 2019-06-19

**Authors:** Dezhong Yu, Xiaofang Liu, Xiao Liu, Wencheng Cao, Xiaotian Zhang, Haoyuan Tian, Jin Wang, Nan Xiong, Sheng Wen, Yongning Wu, Xin Sun, Yan Zhou

**Affiliations:** 1School of Chemical and Environmental Engineering, Wuhan Institute of Technology, LiuFang Campus, No.206, Guanggu 1st road, Wuhan 430205, China; yudezhongwh@163.com (D.Y.); 13971689391@163.com (X.L.); 2Hubei Provincial Key Laboratory for Applied Toxicology, Hubei Provincial Center for Disease Control and Prevention, #6 Zhuo Daoquan North Road, Wuhan 430079, China; xiaoxiao19880924@hotmail.com (X.L.); cwc79279@163.com (W.C.); xtzhang@mail.bnu.edu.cn (X.Z.); wenshenggy@aliyun.com (S.W.); 3Key Laboratory of Chemical Safety and Health, National Institute of Occupational Health and Poison Control, Chinese Center for Disease Control and Prevention, #27 Nan Wei Road, Beijing 100050, China; hywork@163.com (H.T.); jinw1003@hotmail.com (J.W.); 4College of Chemistry and Materials Science, South Central University for Nationalities, #182 Minzu Avenue, Wuhan 430074, China; xiongnan155@126.com; 5The Key Laboratory of Food Safety Risk Assessment, Ministry of Health (CFSA) and China National Center for Food Safety Risk Assessment, #7 Panjiayuan Nanli, Beijing 100021, China; wuyongning@cfsa.net.cn

**Keywords:** PCDD/Fs, dl-PCBs, umbilical cord serum, chemical plant, exposure risk

## Abstract

Polychlorinated dibenzo-p-dioxins, polychlorinated dibenzofurans (PCDD/Fs), and dioxin-like polychlorinated biphenyls (dl-PCBs) are bioaccumulative compounds that may affect fetal growth and infant development. The aim of this study was to determine whether the pregnant women living near a chemical plant in Tianjin had a risk of exposure to dioxins. Concentrations of PCDD/Fs and dl-PCBs in 24 umbilical cord serum samples collected from pregnant women were measured using a high-resolution gas chromatograph with a high-resolution mass spectrometer (HRGC-HRMS) and an isotopic dilution method. The levels of ∑(PCDD/Fs + dl-PCBs) were in the range 476–8307 pg·g^−1^ lipid, with a mean of 3037 pg·g^−1^. The mean World Health Organization toxicity equivalent (WHO-TEQ) for PCDD/Fs and dl-PCBs was 14.0 and 2.14 pg·g^−1^ lipid, respectively. The PCDD/Fs and dl-PCBs contributed 86.7% and 13.3%, respectively, to the total TEQ. The octa-CDFs and penta-CBs were predominant for the PCDD/Fs and dl-PCBs, accounting for 57.6% and 74.3%, respectively. Several PCDD/F and dl-PCB congeners were highly correlated, such as PCB 105 and PCB 118 (r = 0.982, *p* < 0.001). Although the results hint at decreasing trends for PCDD/F and dl-PCBs by comparison with a similar study in Tianjin, a total TEQ of 41.7% of study participants had a body burden that exceeded the biomonitoring equivalents for dioxins. It was shown that pregnant women and infants had a health risk of exposure to dioxins.

## 1. Introduction

Polychlorinated dibenzo-p-dioxins (PCDDs), polychlorinated dibenzofurans (PCDFs), and dioxin-like polychlorinated biphenyls (dl-PCBs) are major classes of persistent organic pollutants (POPs). Owing to their high lipophilicity and persistence, these types of POPs are ubiquitous in the environment. They are shown to induce carcinogenicity, endocrinopathy, neurotoxicity, and immunotoxicity [1,2].

Dioxin appears to have numerous adverse health effects, including growth retardation in fetuses and infants, neurodevelopmental dysfunction, thyroid deficiency, immune deficiency, reproductive effects, and cancer [3,4]. Fetuses and infants are the subpopulation that we must take into consideration when we discuss the effects of PCDD/Fs and PCBs on humans because they tend to be more susceptible to dioxin-like compounds [5]. The importance of health risks in relation to fetal exposure has been underlined by the fact that transplacental transfer of dioxins has been demonstrated by congener specific analyses in human umbilical cord serum, placenta, and fetal tissues in several studies [6,7,8]. One of the most significant concerns regarding health effects is the harmful influence of dioxins on future generations, stemming from prenatal or postnatal exposure. The collection of umbilical cord serum from the placenta is more directly linked to fetal exposure and less invasive than venipuncture of pregnant women [9,10,11,12,13]. Dioxins accumulated in women’s bodies have been found to be passed on to their babies by both transplacental and lactational routes [14].

Tianjin once was a traditional base of the chemical industry in China; facilities there have produced lindane, chlorobenzene, and pentachlorophenol (PCP) for 30–50 years [15,16]. From 1958 to 2004, PCP and its sodium salt (Na-PCP) were manufactured by the Tianjin Dagu Chemical Company. Many studies have reported that PCP production may be a particularly important source of PCDD/Fs [17,18]. Significant exposure to the workers has previously been demonstrated by very high blood levels of PCDDs/PCDFs [19]. Toxicity equivalents (TEQs) of PCDD/Fs and dl-PCBs have also been detected in the serum of child-bearing women in the vicinity of Dagu [20].

In the present work, we conducted a small pilot study to explore the general levels of dioxins in umbilical cord serum of pregnant women living near the Tianjin Dagu Chemical Company. The purpose of this study was to determine the concentrations and the distribution of 17 PCDD/F and 12 dl-PCB congener profiles, to study the trend of change of dioxins in local pregnant women, and to assess the health risk related to fetal exposure.

## 2. Materials and Methods

### 2.1. Sampling

Twenty-four umbilical cord serum samples were collected in 2017 from pregnant women living near the Tianjin Dagu Chemical Company. Before sample collection, all participants completed an informed consent form and an exposure assessment questionnaire. The nurse collected 5 mL of cord blood in vials at birth and centrifuged immediately. The serum samples were then segregated and sent on dry ice to the Dioxin Reference Laboratory (Chinese Ministry of Health) for analysis. The serum samples were frozen and preserved at −40 °C before analysis.

### 2.2. Analytical Methods and Instrumentation

An analysis of PCDD/Fs and dl-PCBs was performed according to United States Environmental Protection Agency (US EPA) methods 1613B [21] and 1668C [22], with some modifications. Briefly, 2 mL serum was mixed with approximately 20 g diatomite and then spiked with ^13^C-labeled surrogate standards (EPA-1613LCS and P48-W-ES). The mixture was extracted in three cycles using an accelerated solvent extractor (ASE-914, BÜCHI, Flawil, Switzerland) with an n-hexane/dichloromethane/acetone mixture (45:45:10, *v*/*v*) as solvent at 130 °C and 100 bar. Ten grams of anhydrous sodium sulfate was added to the extract to remove the water. Then, the extract was concentrated to near 1 mL and subsequently purified using tandem columns of acid silica gel and carbon (F12, CAPE Technologies, South Portland, Maine, USA) under a pressure of 0.1 MPa. Two fractions containing PCDD/Fs and dl-PCBs were concentrated to near dryness and then re-dissolved in approximately 20 μL of nonane. The ^13^C-labeled injection standards for PCDD/Fs (EPA-1613ISS) and dl-PCBs (P48-RS) were added before an instrumental analysis. PCDD/Fs and dl-PCBs were analyzed using a high-resolution gas chromatograph–high-resolution mass spectrometer (HRGC-HRMS, DFS, ThermoFisher Scientific, Waltham, MA, USA) with a DB-5 MS capillary column (60 m × 0.25 mm i.d. × 0.25 μm) using an isotopic dilution method for quantification. 

The 17 congener profiles of 2,3,7,8-substituted PCDD/Fs and 12 congener profiles of dl-PCBs designated by the World Health Organization (WHO) were quantified. The total lipid content in each serum sample was determined in a <0.1 mL aliquot of serum specimen using an enzymatic summation method [23]. The levels of the analytes were adjusted to a lipid weight basis.

### 2.3. Quality Assurance and Quality Control

An isotope dilution method was used to quantify the target compounds. Recovery of ^13^C-labeled surrogates was between 34% and 86% for all samples, which satisfied the EPA methods 1613B and 1668C. The method detection limit calculated (three times the signal-to-noise ratio) was 0.12–0.39 pg·g^−1^ lipid for PCDD/Fs, and 0.55–1.22 pg·g^−1^ lipid for dl-PCBs. To evaluate the accuracy and reliability in the congener-specific analysis of PCDD/Fs and dl-PCBs, standard reference materials (SRM 1957 and SRM 1958, National Institute of Standards and Technology, NIST) were used in our laboratory. Our laboratory also participates in the worldwide annual program of training and accreditation, and in various international proficiency-testing programs such as the Worldwide Interlaboratory Comparison Study on the Determination of POPs organized by the Norwegian Institute of Public Health (NIPH) and United Nations Environment Programme (UNEP). 

### 2.4. Statistic Analysis

All statistical analyses were performed using SPSS version 19.0 (IBM, Armonk, NY, USA). Values below the detection limit were made to be equal to zero for statistical calculations. Toxicity equivalents (TEQs) were calculated using the World Health Organization 2005 toxicity equivalence factors (TEFs) [24].

## 3. Results and Discussion

### 3.1. PCDD/F and dl-PCB Concentrations and Their TEQs

Table 1 presents the distribution of PCDD/Fs and dl-PCBs concentrations in cord serum among the study participants. The detection rate of all compounds ranged from 20.8% to 100%, among which PCB114, PCB118, PCB156, PCB157, PCB167, PCB105, PCB123, 2,3,7,8-TCDF (Tetrachlorodibenzofuran), PCB77, and OCDD (Octachlorodibenzodioxin) showed higher detection frequencies of 100.0%, 100.0%, 100.0%, 100.0%, 100.0%, 95.8%, 95.8%, 95.8%, 91.7%, and 91.7%, respectively. 

The mean concentration of ∑(PCDD/Fs+ dl-PCBs) was 3037 pg·g^−1^ lipid, and the range was 476 to 8307 pg·g^−1^ lipid. For PCDD/Fs, the total concentrations ranged from 43.0 to 1335 pg·g^−1^ lipid, with a mean of 336 pg·g^−1^ lipid. OCDD (mean 183 pg·g^−1^ lipid) was the predominant PCDD/F congener profile, followed by 1,2,3,4,6,7,8-HpCDD (30.9 pg·g^−1^ lipid) and 2,3,7,8-TCDF (22.7 pg·g^−1^ lipid). The predominant dl-PCB was PCB 118 (mean 1311 pg·g^−1^ lipid), followed by PCB 105 (568 pg·g^−1^ lipid) and PCB 156 (265 pg·g^−1^ lipid). The total concentrations ranged from 330 to 7908 pg·g^−1^ lipid, with a mean of 2700 pg·g^−1^ lipid. The mean TEQ for PCDD/F and dl-PCB was 14.0 and 2.14 pg·g^−1^ lipid, respectively. Considering the mean values, the relative contribution of PCDD/Fs and dl-PCBs to total TEQ was 86.7% and 13.3%, respectively.

### 3.2. Congener Profiles of PCDD/Fs and dl-PCBs 

Figure 1 shows the PCDD/F congener profile in cord serum samples from Tianjin. OCDD was predominant, accounting for 54.5%, followed by 1,2,3,4,6,7,8-HpCDD (9.2%) and 2,3,7,8-TCDF (6.8%). Similar patterns were also observed in other studies [14,25]. Dioxins are unwanted contaminants almost exclusively produced by industrial processes, including incineration of municipal solid waste or medicinal waste, cement production, fuel combustion, chlorine bleaching of paper and pulp, and the manufacture of some pesticides, herbicides, and fungicides [26,27,28]. Tianjin city, which is located in northern China, was once a major organochlorine pesticide production area, where most of the industries related to chemical pesticides were concentrated [29,30]. Members of the PCDD/F congener groups are primarily composed of octa-CDD/Fs, accounting for 57.6%, as shown in Figure 2. From 1958 to 2004, PCP and PCP-Na were manufactured in large quantities by the Dagu Chemical Company in Tianjin. Bao et al. [18] reported that PCP and Na-PCP contain large amounts of dioxins, and octa-CDD/Fs were predominant in 17 PCDD/Fs. Similar congener profiles of PCDD/Fs in sediments from Tianjin have also been reported in other studies [31,32,33].

With regard to dl-PCBs, nine out of twelve congeners showed high rates of detection (more than 90%) in all cord serum samples. The most abundant congener was PCB 118, accounting for 48.6%, followed by PCB 105 (21.0%) and PCB 156 (9.8%), as shown in Figure 3. Figure 4 shows that pentachlorobiphenyls (penta-CBs) were the main PCB congeners in cord serum samples from Tianjin, which accounted for 74.3%, followed by hexachlorobiphenyls (tetra-CBs, 16.2%), tetrachlorobiphenyls (hexa-CBs, 8.1%), and heptachlorobiphenyls (hepta-CBs, 1.4%). PCBs were produced in China from 1965, and the main products were tri-CBs and penta-CBs [34]. Tianjin once was a traditional base of the chemical industry in China, where PCP was produced for 30–50 years. Similar PCB congener groups were also reported in other cities of China [35].

Significant correlations existed among concentrations of certain PCDD/F and dl-PCB congeners in cord serum, which were examined using Spearman’s correlation coefficient tests. The results suggested the exposure sources and kinetics of the contaminants. Only the significant correlations are listed in Table 2. The highest correlation was observed between PCB 105 and PCB 118 (r = 0.982, *p* < 0.001). We found that OCDD showed positive correlations with all penta-CBs of dl-PCBs. This indicated that they may have a similar exposure source. Because Tianjin was once a traditional base of chemical industry in China, pentachlorophenol and its sodium salt (PCP and Na-PCP) were manufactured in Tianjin Dagu Chemical Company from 1958 to 2004. Many studies have reported that PCP and Na-PCP production could be a very important source of OCDD and penta-CBs [17,18,26].

### 3.3. Comparison of TEQs of PCDD/Fs and dl-PCBs with Those in Other Countries

There are many biomonitoring studies of dioxins, such as in human serum [36], mother serum [37], cord blood [38,39], and so on, but only a few studies focused on umbilical cord serum. Wang et al. reported the mean TEQ of 13.56 pg·g^−1^ lipid in cord serum of pregnant women in Taiwan [12]. Hsu et al. studied the general population of Taiwan with mean values of 15.0 pg·g^−1^ lipid [40]. The mean concentration of TEQ ∑(PCDD/Fs + dl-PCBs) was 16.10 pg·g^−1^ lipid in cord serum samples in our studies, which is higher than the TEQ reference background level of non-exposed adults in the general populations (mean value of 13.2 pg·g^−1^ lipid) in a comprehensive worldwide literature review and in other studies [41].

For PCDD/Fs, the mean TEQ of 13.96 pg·g^−1^ lipid in our study was lower than the TEQ (mean 14.5 pg·g^−1^ lipid) in cord serum in a Vietnamese study reported by Boda et al. [42]. Our observations also indicate that the TEQ of dl-PCBs (mean 2.14 pg·g^−1^ lipid) in cord serum was similar to what was found in a Japanese study (mean 2.4 pg·g^−1^ lipid [8]). TEQ dl-PCBs contributed only about 13.3% to the total TEQ value. First, the total amount of PCBs produced in China accounted for a small proportion (0.6%) of the amount produced globally [43]. Second, from the 1970s, the Chinese government increasingly regulated and controlled the production, importation, use, treatment, and disposal of PCBs and increasingly managed the production and quality of commercially produced foodstuffs. Moreover, the observed TEQ concentrations of dl-PCBs may be due to the general background exposure in the city because all of the groups exhibited TEQs of dl-PCBs equal to or lower than those for other environmentally or occupationally exposed populations reported in the literature [44,45].

Meanwhile, Suzuki, et al. [8] reported that the TEQ of maternal blood (mean 35.3 pg·g^−1^ lipid) was significantly higher than the TEQ of cord serum (16.3 pg·g^−1^ lipid) in Japan. They concluded that high TEQ accumulated in the placenta when dioxins were transferred from the mother to fetus. Chen et al. [20] reported that childbearing age women living close to the Tianjin Dagu Chemical Company in 2013 had a mean TEQ of 62.5 pg·g^−1^ lipid, which is significantly higher than in our research in 2017. Although they are not the same people, they are all participants living near the local chemical plant (Dagu). The results indicate that pollutants may be adsorbed or obstructed by placentas during delivery. More likely, the pollutions may decrease as time goes on with rising effectiveness of legislatively banned use of organochlorine pesticides in Tianjin.

### 3.4. Risk Assessment

For risk assessment purposes, the results can also be compared with biomonitoring equivalents (BEs). These values are defined as the concentration, or range of concentrations, of a chemical in blood that is consistent with an existing health-based exposure guideline, such as reference doses. BE values are designed to be used as screening tools to assess whether chemicals have a large, a small, or no margin of safety compared to existing health-based exposure guidelines [46]. Aylward et al. reported that a serum lipid-adjusted TEQ of approximately 15 pg·g^−1^ lipid, on the basis of neurodevelopmental effects, is consistent with the minimal risk level (MRL) recommended by the Agency for Toxic Substances and Disease Registry [47]. Referring to the TEQs ∑(PCDDs/Fs + dl-PCBs) in our data, 42.7% of the participants exhibited levels higher than the BE value. This result indicates that there was an exposure risk of dioxin to pregnant women and infants, and that further efforts to monitor dioxins are necessary.

## 4. Conclusions

In summary, the present study showed the existence of areas with potentially higher exposure of dioxins to pregnant women residing in the vicinity of a chemical plant. The octa-CDFs and penta-CBs were predominant for the PCDD/Fs and dl-PCBs, respectively. Several PCDD/F and dl-PCB congeners were highly correlated. The decreasing concentration of TEQs ∑(PCDD/Fs + dl-PCBs) in this study may indicate that the legislative banning of the use of organochlorine pesticides in Tianjin was effective and that the pollutants were possibly adsorbed or obstructed by placentas during delivery. However, monitoring of prenatal exposure of dioxins is still necessary due to their persistence and toxicology. The positive correlation among some targeted compounds suggested a similar exposure source. The results of this study provide basic information for monitoring and pollution control of PCDD/Fs and dl-PCBs in chemical industrial districts. However, it is necessary to quantify the concentrations of these pollutants again in the future in order to reveal the relationship between POPs and pregnancy outcomes.

## Figures and Tables

**Figure 1 ijerph-16-02178-f001:**
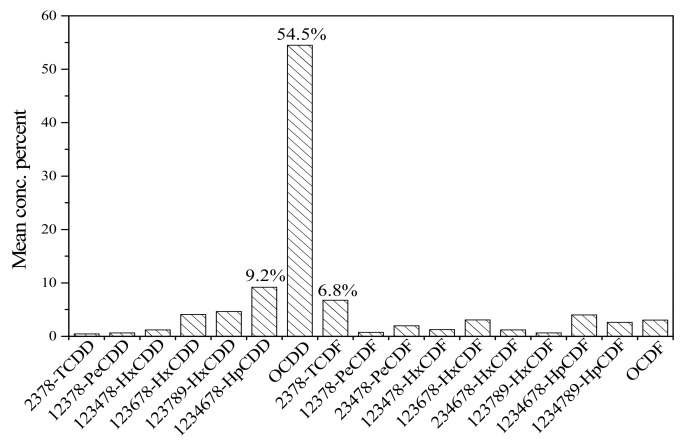
PCDD/F congener profiles in cord serum samples.

**Figure 2 ijerph-16-02178-f002:**
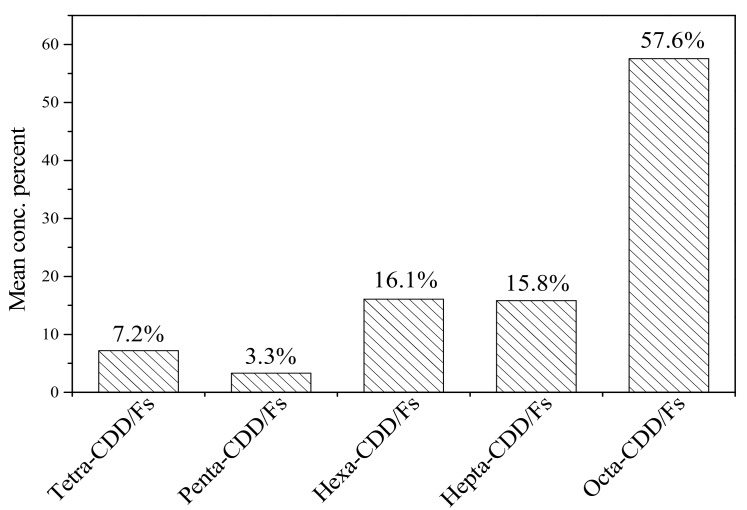
Comparison of different PCDD/F congener groups in cord serum samples.

**Figure 3 ijerph-16-02178-f003:**
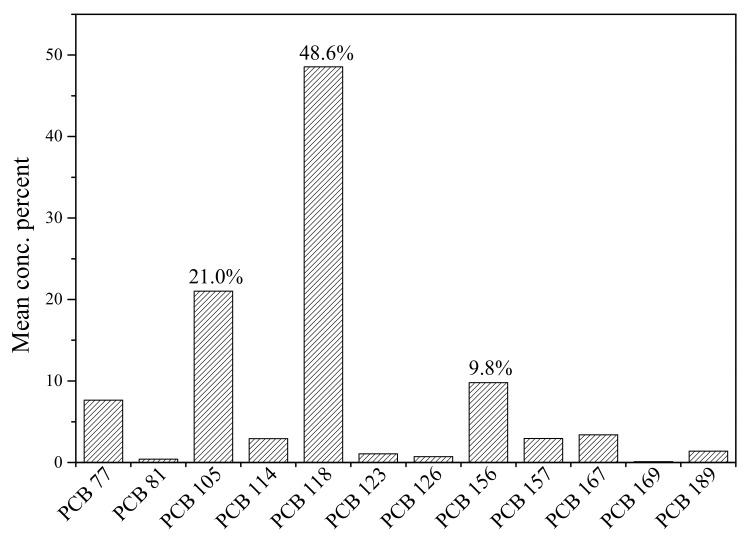
dl-PCB congener profiles in cord serum samples.

**Figure 4 ijerph-16-02178-f004:**
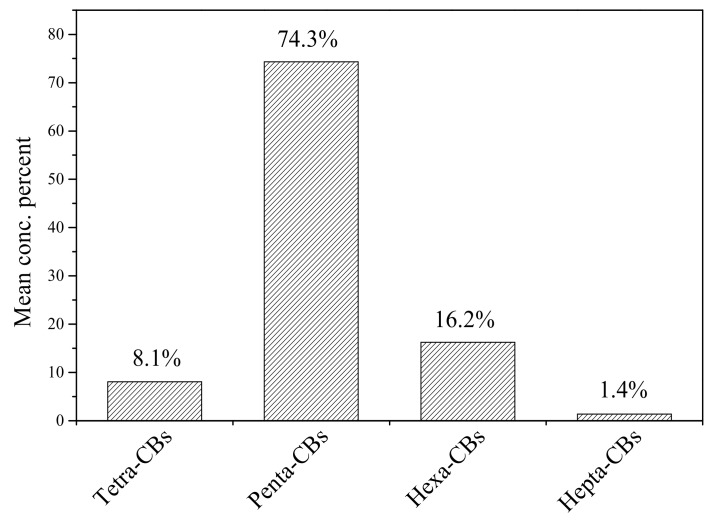
Comparison of different dl-PCB congener groups in the cord serum samples.

**Table 1 ijerph-16-02178-t001:** Descriptive statistics of PCDD/Fs and dl-PCBs congers in cord serum samples, expressed as pg·g^−1^ lipid (*n* = 24).

Compounds	DL	DR (%)	Min.	Median	Max.	Mean	Mean TEQ
∑PCDD/Fs		100.0	43.0	227	1335	336	14.0
∑PCDDs		100.0	9.76	175	1234	252	7.34
2,3,7,8-TCDD	0.24	20.8	ND	ND	14.0	1.49	1.49
1,2,3,7,8-PeCDD	0.36	36.3	ND	ND	18.7	2.14	2.14
1,2,3,4,7,8-HxCDD	0.12	37.5	ND	ND	33.8	4.08	0.41
1,2,3,6,7,8-HxCDD	0.12	62.5	ND	3.99	164	13.8	1.38
1,2,3,7,8,9-HxCDD	0.12	54.2	ND	2.95	150	15.6	1.56
1,2,3,4,6,7,8-HpCDD	0.24	70.8	ND	5.28	470	30.9	0.31
OCDD	0.20	91.7	ND	141	1200	183	0.06
∑PCDFs		100.0	17.6	48.5	296	85.1	6.62
2,3,7,8-TCDF	0.20	95.8	ND	16.4	75.5	22.7	2.27
1,2,3,7,8-PeCDF	0.39	33,3	ND	ND	35.8	2.45	0.07
2,3,4,7,8-PeCDF	0.36	62.5	ND	2.18	46.8	6.60	1.98
1,2,3,4,7,8-HxCDF	0.24	54.2	ND	2.20	13.9	4.23	0.42
1,2,3,6,7,8-HxCDF	0.20	70.8	ND	4.05	111	10.4	1.04
2,3,4,6,7,8-HxCDF	0.16	66.7	ND	1.03	27.8	3.98	0.40
1,2,3,7,8,9-HxCDF	0.20	50.0	ND	ND	17.9	2.13	0.21
1,2,3,4,6,7,8-HpCDF	0.20	29.2	ND	3.99	111	13.5	0.14
1,2,3,4,7,8,9-HpCDF	0.24	75.0	ND	ND	171	8.84	0.09
OCDF	0.16	29.2	ND	ND	108	10.2	0.003
∑dl-PCBs		100.0	330	1323	7908	2700	2.14
∑non-ortho PCBs		100.0	6.10	42.7	963	240	2.06
PCB 77	0.63	91.7	ND	33.2	880	207	0.02
PCB 81	0.55	58.3	ND	5.34	35.5	11.1	0.003
PCB 126	0.87	66.7	ND	16.1	75.0	19.6	1.96
PCB 169	1.18	37.5	ND	ND	17.9	2.72	0.08
∑mono-ortho PCBs		100.0	303	1226	6944	2460	0.07
PCB 105	0.95	95.8	ND	170	2045	568.	0.02
PCB 114	0.67	100.0	10.6	49.6	225	79.3	0.002
PCB 118	0.75	100.0	138	585	3778	1311	0.04
PCB 123	0.79	95.8	ND	18.7	94.8	28.8	0.0009
PCB 156	1.18	100.0	31.0	201	604	265	0.08
PCB 157	1.18	100.0	17.6	69.1	217	79.5	0.002
PCB 167	1.07	100.0	13.0	66.9	217	91.3	0.003
PCB 189	1.22	79.2	ND	19.4	113	37.8	0.001
∑(PCDD/Fs + dl-PCBs)		100.0	476	1842	8307	3037	16.1

DL, detection limit; DR, detection rate; ND, not detected; TEQ: Toxicity equivalent; PCDD/Fs: Polychlorinated Dibenzo-p-Dioxins and Polychlorinated Dibenzofurans; dl-PCBs: Dioxin-like Polychlorinated Biphenyls; TCDD: Tetrachlorodibenzo-p-dioxin; PeCDD: Pentachlorodibenzo-p-dioxin; HxCDD: Hexachlorodibenzo-p-dioxin; HpCDD: Heptachlorodibenzo-p-dioxin; OCDD: Octachlorodibenzodioxin; TCDF: Tetrachlorodibenzofuran; PeCDF: Hexachlorodibenzofuran; HxCDF: Hexachlorodibenzofuran; HpCDF: Heptachlorodibenzofuran; OCDF: Octachlorodibenzofuran.

**Table 2 ijerph-16-02178-t002:** Spearman’s correlation coefficients among concentrations of PCDD/Fs and dl-PCBs.

	OCDD	PCB 105	PCB 114	PCB 118	PCB 123	PCB 126
OCDD	1					
PCB 105	0.61 **	1				
PCB 114	0.52 **	0.89 ***	1			
PCB 118	0.56 **	0.98 ***	0.90 ***	1		
PCB 123	0.54 **	0.72 ***	0.74 ***	0.74 ***	1	
PCB 126	0.59 **	0.50 *	0.48 *	0.48 *	0.41*	1

* *p* < 0.05; ** *p* < 0.01; *** *p* < 0.001.

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
