# Peer review of "Polychlorinated Dibenzo-p-Dioxins, Polychlorinated Dibenzofurans, and Dioxin-Like Polychlorinated Biphenyls in Umbilical Cord Serum from Pregnant Women Living Near a Chemical Plant in Tianjin, China"

_ijerph, 2019, doi:10.3390/ijerph16122178_

Round 1

Reviewer 1 Report

The manuscript is well elaborated and easy to follow. I have only few comments:

Line 27: ‘The test results, …’ it should be rephrased.

Lines 35-36: The final conclusion: ‘It was shown that pregnant women and infants had a risk of exposure to dioxins.’ should be rephrased. Exposure to dioxin is common around the world. Authors assessed the health risk and they should comment on this not the risk of exposure.

Paragraph 2.1 According to the description, authors used plasma not serum in this study. Collection of blood with EDTA tubes allows to separate plasma. It should be explained.

Line 135. Table 1. It is not informative to use arithmetic mean when detection ratio is lower than 50-60%. It is advisable to show selected percentiles to describe distribution of concentrations.

Author Response

Point 1: The test results, …’ it should be rephrased.

Response 1: Thanks for your suggestion. We have rephrased it in the paper. “The levels of ∑(PCDD/Fs+ dl-PCBs) were in the range 476–8307 pg g-1 lipid, with a mean of 3037 pg g-1.”

Point 2: The final conclusion: ‘It was shown that pregnant women and infants had a risk of exposure to dioxins.’ should be rephrased. Exposure to dioxin is common around the world. Authors assessed the health risk and they should comment on this not the risk of exposure.

Response 2: Thanks for your suggestion. It has been modified in the paper. “It was shown that pregnant women and infants had a health risk of exposure to dioxins.”

Point 3: Paragraph 2.1 According to the description, authors used plasma not serum in this study. Collection of blood with EDTA tubes allows to separate plasma. It should be explained.

Response 3: Thanks for your suggestion. It has been modified in the paper. “The nurse collected five milliliters of cord blood vials at birth and centrifuged immediately. Then the serum samples were segregated and sent on dry ice to the Dioxin Reference Laboratory (Chinese Ministry of Health) for analysis.”

Point 4: Table 1. It is not informative to use arithmetic mean when detection ratio is lower than 50-60%. It is advisable to show selected percentiles to describe distribution of concentrations.

Response 4: Thanks for your suggestion. Because the detection ration of seven compounds are below 50% of the thirty-six compounds in Table 1, so we uniform the distribution pattern and this pattern is useful for our further analysis.

Reviewer 2 Report

Title: Polychlorinated dibenzo-p-dioxins, polychlorinated dibenzofurans, and dioxin-like polychlorinated biphenyls in umbilical cord serum from pregnant women living near a chemical plant in Tianjin, China

This paper is well done, the study is very interesting, but it is very simple: few cord human samples can give us just a trend of exposure to PCDDs/Fs and DL-PCBs. It would be interesting also to know levels of NDL-PCBs: why do you not research them? Their chemical analysis is simpler than PCDD/F and DL-PCB analysis and you can do it collecting few mL of serum (about 5-6). Then why did you not collect other important information? For example, data on anthropometric and socio-demographic characteristics, diet, and so on, would be considered and correlated with results to have other important evidence. It’s true that, maybe, 24 samples are really few, but you could try!

So this paper can be considered just the beginning of a more accurate and interesting study. Then I suggest you have another bibliography research, introducing more recent papers and removing older ones.  

Reported PCDD/F data are very high for a cord serum sample!!! These data seem to define a health emergency. So I think that other biomonitoring general population studies are necessary and more legislative action will be needful in the next future.

Some clarifications are necessary:

Line 84: please, write US EPA method 1613B 1994 and not 1997

Lines 84 and 105: for your information, EPA US EPA method 1668A 1999 is old, method 1668C 2010 is available.

Lines 111-112: you write about your participation in Ring Test (NIPH and UNEP). I can’t understand in which matrices POPS are determined: in human serum samples? Also does Norwegian Institute Public Health organize PT on serum? I didn’t know it. If it can be interesting to you, a good Canadian Ring Test on human serum is available https://www.inspq.qc.ca/en/ctq/eqas/amap/description).

Lines 120-124: reading is not simple, maybe a table can help reader or you can reduce.

Line 186: do you mean biomonitoring studies in China? Otherwise I don’t agree with you when you write “few studies”. In the world there are many biomonitoring studies: human serum, mother serum, cord serum, human milk, placenta and so on. Please, see recent studies and rewrite the 3.3 paragraph.

Line 228-232: you don’t have a scientific evidence, because you didn’t collect mother’s serum samples, you have only cord serum samples. What you say can be deduced, also if even literature confirms. For this it is very important you refer to recent studies in your paper.  Why did you not collect mother’s serum? During pregnant women are used to blood tests.

However, you have to rewrite the “Conclusion” paragraph indicating results obtained in your study.

Finally, several times in the paper you write about the industrial origin of Tianjin and PCP production: please, summarize and report only once.

Author Response

Response to Reviewer 2 Comments

Point 1: please, write US EPA method 1613B 1994 and not 1997

Response 1: Thanks for your suggestion. I have found the standard method of 1994 and It has been modified in the paper. “U.S. Environmental Protection Agency (EPA). Method 1613 Revision B: Tetra-through octachlorinated dioxins and furans by isotope dilution HRGC/HRMS, EPA 621-B-94-005. United States Environmental Protection Agency Office of Water: Washington, DC, 1994.”

Point 2: for your information, EPA US EPA method 1668A 1999 is old, method 1668C 2010 is available.

Response 2: Thanks for your suggestion. I have found the latest standard method and It has been modified in the paper. “U.S. Environmental Protection Agency (EPA). Method 1668, Revision C: Chlorinated biphenyl congeners in water, soil, sediment, biosolids and tissue by HRGC/HRMS, EPA820-R-10-005. Environmental Protection Agency Office of Water: Washington, DC, 2010.”

Point 3: you write about your participation in Ring Test (NIPH and UNEP). I can’t understand in which matrices POPS are determined: in human serum samples? Also does Norwegian Institute Public Health organize PT on serum? I didn’t know it. If it can be interesting to you, a good Canadian Ring Test on human serum is available in human serum samples?https://www.inspq.qc.ca/en/ctq/eqas/amap/description).

Response 3: Thanks for your suggestion. The matrices of the two programmes include human milk, food and feed, sediment, and so on, but not serum samples until now. We write it to prove our analytical ability. Thank you very much for the suggestion of Canadian Ring Test on human serum.

Point 4: reading is not simple, maybe a table can help reader or you can reduce.

Response 4: Thanks for your suggestion. Table 1 have shown all detection rates.

Point 5: do you mean biomonitoring studies in China? Otherwise I don’t agree with you when you write “few studies”. In the world there are many biomonitoring studies: human serum, mother serum, cord serum, human milk, placenta and so on. Please, see recent studies and rewrite the 3.3 paragraph.

Response 5: Thanks for your suggestion. There are many biomonitoring studies: human serum, mother serum, cord blood and so on. In this paper we mainly studied the dioxins in umbilical cord serum and the serum of non-exposed people. It is convenient for us to compare the levels of dioxins in the same matrix of umbilical cord serum, so We have rewrite the paragraph in the paper. “There are many biomonitoring studies of dioxins, such as human serum, mother serum, cord blood and so on, but only a few studies focused on umbilical cord serum.”

Point 6: you don’t have a scientific evidence, because you didn’t collect mother’s serum samples, you have only cord serum samples. What you say can be deduced, also if even literature confirms. For this it is very important you refer to recent studies in your paper. Why did you not collect mother’s serum? During pregnant women are used to blood tests.

Response 6: Thanks for your suggestion. Because this is a pilot experiment, we will monitor mother’s serum, cord serum and breast milk in the future study and to confirm the deduce progress.

Point 7: However, you have to rewrite the “Conclusion” paragraph indicating results obtained in your study.

Response 7: Thanks for your suggestion. We have rewrite the “Conclusion” in the paper. The content is blow: “In summary, the present study showed the existence of areas with potentially higher exposure of dioxins to pregnant women residing in the vicinity of a chemical plant. The octa-CDFs and penta-CBs were predominant for the PCDD/Fs and dl-PCBs, respectively. Several PCDD/F and dl-PCB congeners were highly correlated. The decreasing concentration of TEQs ∑(PCDD/Fs+ dl-PCBs) in this study may indicate that the legislative banning of the use of organochlorine pesticides in Tianjin was effective and maybe the pollutants were adsorbed or obstructed by placentas during delivery. However, monitoring of prenatal exposure of dioxins is still necessary due to their persistence and toxicology. The positive correlation among some targeted compounds suggested a similar exposure source. The results of this study provide basic information for monitoring and pollution control of PCDD/Fs and dl-PCBs in chemical industrial districts. However, it is necessary to quantify the concentrations of these pollutants again in the future in order to reveal the relationship between POPs and pregnancy outcomes.”
